# Evaluation of Fracturing Effect of Tight Reservoirs Based on Deep Learning

**DOI:** 10.3390/s24175775

**Published:** 2024-09-05

**Authors:** Ankang Feng, Yuxin Ke, Chuang Hei

**Affiliations:** School of Electronic Information and Electrical Engineering, Yangtze University, Jingzhou 434023, China; 201606279@yangtzeu.edu.cn (A.F.); keenh6728@gmail.com (Y.K.)

**Keywords:** fracturing prediction, deep learning, generative adversarial network, CNN

## Abstract

The utilization of hydraulic fracturing technology is indispensable for unlocking the potential of tight oil and gas reservoirs. Understanding and accurately evaluating the impact of fracturing is pivotal in maximizing oil and gas production and optimizing wellbore performance. Currently, evaluation methods based on acoustic logging, such as orthogonal dipole anisotropy and radial tomography imaging, are widely used. However, when the fractures generated by hydraulic fracturing form a network-like pattern, orthogonal dipole anisotropy fails to accurately assess the fracturing effects. Radial tomography imaging can address this issue, but it is challenged by high manpower and time costs. This study aims to develop a more efficient and accurate method for evaluating fracturing effects in tight reservoirs using deep learning techniques. Specifically, the method utilizes dipole array acoustic logging curves recorded before and after fracturing. Manual labeling was conducted by integrating logging data interpretation results. An improved WGAN-GP was employed to generate adversarial samples for data augmentation, and fracturing effect evaluation was implemented using SE-ResNet, ResNet, and DenseNet. The experimental results demonstrated that ResNet with residual connections is more suitable for the dataset in this study, achieving higher accuracy in fracturing effect evaluation. The inclusion of the SE module further enhanced model accuracy by adaptively adjusting the weights of feature map channels, with the highest accuracy reaching 99.75%.

## 1. Introduction

In the development of tight reservoirs, characterized by low permeability and high density, hydraulic fracturing technology is commonly employed to enhance oil and gas production [1,2]. The fracturing effect of the reservoir directly influences the subsequent oil and gas development process. The fractured zone forms an elliptical area along the wellbore, and evaluating the fracture effectiveness in the vertical (z-direction) height and radial (x-direction) dimensions is a key factor. Hence, after applying fracturing technology in unconventional reservoirs, an effective method is needed to accurately evaluate the fracturing effect of the reservoir.

Conventional evaluation methods are typically based on manual observation or manual calculation, which are susceptible to human factors and require extensive human resources and time for manual data processing, resulting in high costs and time consumption. The traditional methods mainly include anisotropy inversion technology [3], micro-seismic technology [4], and post-compression data analysis methods [5]. However, these traditional methods have certain limitations. For example, anisotropy inversion technology is influenced by the morphology and orientation of the fracturing fissures. It is limited in its ability to represent the fracturing fissure situation away from the wellbore, making it difficult to evaluate the effectiveness of offshore fracturing operations. Micro-seismic technology evaluates the dynamic formation process of fracturing fissures and requires monitoring wells, significantly increasing the cost of exploration and development, especially for offshore, unconventional oil and gas. The post-compression data analysis method is affected by the fracturing data, resulting in less accurate results in the later evaluation of fracturing effectiveness.

With the development of machine learning, researchers both domestically and internationally have conducted significant work in log curve recognition using machine learning methods [6]. Substantial progress has been made in the recognition, prediction, and completion of log curves [7,8]. Nevertheless, machine learning still faces challenges in learning the underlying rules of complex data and often struggles to achieve strong expressive power and high prediction accuracy when dealing with intricate data classification tasks.

Deep learning is a popular research area in machine learning, which involves using large amounts of data to learn low-level features through deep structural learning and map them to high-level features in order to complete complex classification tasks by utilizing internal patterns in the data. In recent years, deep learning theory has continuously developed, and many scholars have begun to study the use of deep neural networks to process reservoir fracture data, which has also shown excellent performance in complex well-logging data processing applications [9,10,11,12,13]. Convolutional neural networks (CNN) have the ability to extract features and generalize [14,15] and can explore the nonlinear relationship between acoustic logging data and fracturing effect prediction without human intervention, making fracturing effect evaluation more objective [16]. Due to the high complexity of array acoustic logging signals, complex curve shapes, and interference information such as noise in the data, if shallow convolutional neural networks are used, the model can only learn shallow features, and it is difficult to achieve correct evaluation of fracturing effects [17]. In contrast, deep networks can extract different features and abstract representations and combine these features to form higher-level concepts and representations [18]. However, designing a network that is too deep can lead to the problems of vanishing gradients or exploding gradients, making the model prone to overfitting [19]. 

The dipole array acoustic logging dataset is limited and challenging to collect. Therefore, this study employed data augmentation methods to expand the dataset. In deep learning, data augmentation is a crucial technique that enlarges the original dataset, leading to an increased amount of training data and improved model training performance. Additionally, data augmentation can also alleviate overfitting, enhance the model’s generalization ability and robustness, improve the model’s learning ability on the data, and resist slight perturbations [20]. The most widely used approach for data augmentation is the Generative Adversarial Networks (GANs) proposed by Goodfellow et al. in 2014 [21]. However, the training metrics of GANs and Deep Convolutional GANs (DCGANs) such as KL and JS divergence [22,23] are designed to deal with continuous data, and their performance on discrete data is not ideal. The WGAN-GP proposed by Arjovsky et al. [24] uses a smoother Wasserstein distance to measure the distance between two probability distributions and adds a gradient penalty strategy to make it possible to generate discrete data.

The evaluation of fracturing effects in tight reservoirs is essentially a classification task aimed at assessing the presence of fractured regions along the Z-axis of the wellbore. This paper investigates the evaluation method for fracturing effects in tight reservoirs based on deep learning classification models. The feasibility of three commonly used classification models for evaluating fracturing effects was examined, and the diversity of the dataset was enhanced by generating data using an improved WGAN-GP. This further improved the performance of the model.

## 2. Materials and Methods

### 2.1. ResNet Architecture

In traditional deep neural networks, as the depth of the network increases, the performance of the network often saturates or decreases. This is due to the phenomenon of vanishing or exploding gradients, which makes it difficult for the model to converge. ResNet [25] avoids this phenomenon by introducing residual connections in the network. Residual connections refer to adding the output of some layers in the network to the input of subsequent layers, allowing information to be directly transmitted from shallow layers to deep layers. This connection method helps prevent information from disappearing in the network and makes it easier for the network to learn identity mappings. In addition, the deep network structure improves the performance and feature learning ability of the model [26].

The basic module of ResNet is the “residual block”, which consists of two convolutional layers and a skip connection. It can be stacked to form a deep network. In practice, ResNet’s basic module includes two types of residual blocks: BasicBlock and Bottleneck. BasicBlock usually appears in shallower ResNets such as ResNet18 and ResNet34, while Bottleneck is used in deeper ResNets. In this study, we used BasicBlock, which is applied to shallower layers, and its basic module is shown in Figure 1.

Where X is the input feature map, F(X) is the output before the second activation function, and the final output of the residual block is σ(F(X)+X), where σ is the ReLU activation function, and identity is responsible for transmitting low-level feature information.

Considering the small size of the dataset and with some noise, the ResNet model structure in this paper mainly consists of three residual blocks. The ResNet architecture used in this paper is shown in Figure 2.

### 2.2. SE_ResNet Architecture

The SE module is a module used to enhance the expressive power of deep neural networks. It can adaptively adjust the channel weights of feature maps and more accurately capture critical feature information on the basis of deep networks [27]. As an attention mechanism, the SE module, due to its flexibility, can be embedded into various types of neural networks and is typically used in convolutional neural networks and residual networks. It mainly consists of two operations: the Squeeze operation and Excitation operation. The Squeeze operation reduces the dimension of the feature map by using global average pooling, compressing the information of each channel into a single value. The Excitation operation, similar to the gating units in recurrent neural networks (RNNs), learns the importance weight of each channel through two fully connected layers and weights and sums the values of each channel to obtain a new feature map.

In SE_ResNet, the SE module is incorporated after the residual block structure, as illustrated in Figure 3. The “Residual” in the figure refers to the two convolutional layers before the addition of the feature maps in the residual block. The SE module’s role here is to further enhance the network’s ability to focus on the most relevant features, thereby improving overall model performance.

As shown in Figure 4, the SE_ResNet model used in this paper consists of three modules, where the first module is the Basic Block, and the second and third modules are SE_BasicBlock. The left figure in the following diagram illustrates the basic structure of SE_ResNet, while the right figure illustrates the basic structure of SE_BasicBlock.

### 2.3. DenseNet Structure

Unlike ResNet, which uses residual connections to address the problem of gradient vanishing, DenseNet [28] uses dense connections to pass the feature maps of each layer as inputs to subsequent convolutional layers, thus improving parameter sharing and reducing the number of parameters in the model. DenseNet uses 1x1 convolutional layers to reduce the computational burden of dense connections and improve training efficiency. Additionally, the network structure of DenseNet is quite flexible, allowing it to adapt to different data and tasks by using different connection patterns and layer depths. Its main modules include DenseBlock and Transition. Due to these features, DenseNet can also train deeper networks and achieve good performance on complex well-logging datasets.

One of the core modules of DenseNet is the Dense Block, which consists of multiple densely connected convolutional layers that are directly connected to all preceding layers. This connection allows the feature maps to flow freely throughout the network, effectively alleviating the problem of vanishing gradients and improving the reuse of features and information flow efficiency. Assuming the input to the Dense Block is x_0, the output x_dl of the l-th layer in the dense connection block can be represented as follows:(1)$$x_{dl}=H_{l}\left[x_{0},x_{1},\ldots ,x_{l-1}\right]$$
where H_l_ represents the nonlinear transformation function, which includes a series of operations of BN, ReLU, 1 × 1 convolution kernel (with stride 1), BN, ReLU, and 3 × 3 convolution kernel (with stride 1). The concatenation [x_0_, x_1_, …, x_l−1_] combines the outputs from layers 0 to l−1 in the network. Transition layers are used to connect different Dense blocks, reducing the size of the feature layer and compressing the model. The main operations of the transition layer include BN, ReLU, 1 × 1 convolution kernel (with a stride of 1), dropout, and maximum pooling change. Its output can be expressed as Equation (2):(2)$$x_{trans}=H_{trans}(x_{dl})$$

Due to the dense connection property of the DenseNet model, it is more suitable for large datasets. When used on small datasets, the dense connections can easily lead to overfitting. Therefore, the design of DenseNet should not be too deep. In this study, only two DenseBlocks and one Transition were used, as shown in Figure 5.

### 2.4. Theory and Structure of WGAN-GP

WGAN-GP through the interaction of two deep neural networks: the generator network and the discriminator network. Specifically, the generator network G takes a random distribution Z as input and learns the real data distribution *Pr* to generate data x. The discriminator network D estimates the probability *p* that the input data x are real, distinguishing between the real data distribution and the generated data distribution. The *p*-value reflects the distance between these two distributions. Through continuous adversarial training, both networks improve their respective capabilities—generative and discriminative—until the discriminator is unable to distinguish between the generated data and the real data, resulting in a probability of 0.5 when determining the origin of the input data. At this equilibrium, the generator produces data that are almost indistinguishable from the real data distribution. Figure 6 illustrates the overall architecture of WGAN-GP.

The original Generative Adversarial Networks (GANs) experience slow convergence and gradient vanishing issues. To address these problems, Gulrajani et al. [29] introduced the Wasserstein distance as a loss function, which improves GAN training by optimizing the output distribution of the discriminator. By adding a gradient penalty based on the Lipschitz constraint to the Wasserstein distance, the model gains increased stability during training.

The gradient penalty is calculated as follows: A real sample *x_r_* is randomly drawn from the real data distribution P_r_, while a generated sample x_g_ is taken from the generated data distribution P_g_. Additionally, a random number ∂ is sampled from a uniform distribution on [0,1]. The interpolated sample $\hat{x}$ is then computed using the expression provided in Equation (3):(3)$$\hat{x}=\partial x_{r}+\left(1-\partial \right)x_{g}$$

In training, the term (‖∇*xD*(*x*)‖−1)2 is introduced as a penalty to enforce the gradient norm ‖∇*xD*(*x*)‖ to approximate 1. This penalty is crucial for stabilizing the training process. Consequently, the optimization objective of the improved WGAN network is expressed in Equation (4):(4)$$W\left(P_{r},P_{g}\right)=-{max}_{D}\left\{E_{x\sim P_{r}}\left[D\left(x\right)\right]-E_{x\sim P_{g}}\left[D\left(x\right)\right]-\alpha E_{x\sim P_{\hat{x}}}\left[{\left(\left\|\nabla xD\left(x\right)\right\|-1\right)}^{2}\right]\right\}$$

$E_{x\sim P_{g}}$ and $E_{x\sim P_{r}}$ represent the expected values of the generated and real samples, respectively. The term $P_{\hat{x}}$ denotes the distribution of the interpolated sample $\hat{x}$, with $E_{x\sim P_{\hat{x}}}$ as its expected value. The function *D*(*x*) is the output of the discriminator, and α is the penalty coefficient [30].

To generate more diverse and suitable signal samples, this study introduced several enhancements to the original WGAN. The generator network structure is shown in Figure 7a. First, the negative amplitudes of the generated signals were preserved, avoiding truncation, and the final activation function layer in the generator’s upsampling stage was removed. To further enhance the diversity of the generated signals, Leaky ReLU was employed as the activation function, allowing negative values to be retained. The expression for Leaky ReLU is provided in Equation (5):(5)$$y_{i}=\{\begin{array}{c} x_{i}\;,x_{i}\geq 0\\ ax_{i}\;,x_{i}<0 \end{array}$$
where a is a small decimal approaching zero. After parameter tuning, it was set to 0.1. 

In addition, to preserve the effectiveness of the Lipschitz continuity constraint, all normalization layers were removed from the discriminator to ensure consistency in gradient calculations. The Sigmoid layer was replaced with a 1x1 convolutional layer to avoid the limitations of the Sigmoid function, thereby retaining more detail and signal characteristics while integrating features across different channels. Leaky ReLU was used as the activation function in the discriminator, consistent with the generator. The structure of the discriminator is illustrated in Figure 7b.

## 3. Experimental Design

Combined with the overall workflow in Figure 8 and the proposed network model in this study, the experimental steps were as follows:Dataset preprocessing: In this study, the sonic logging data before and after the hydraulic fracturing of the tight reservoir layer were inconsistent. Therefore, the sonic logging data needed to be truncated to the same depth range, and band-pass filtering and normalization processing were performed on the truncated data. Meanwhile, the hydraulic fracturing section was manually labeled as 1, and the non-fracturing section was labeled as 0 based on the comprehensive processing interpretation results of the sonic logging data;WGAN-GP-generated signals: An improved WGAN-GP was used to generate adversarial samples, and the generated sample set was filtered and normalized;Hyperparameter optimization: A controlled variable method was used to fine-tune the parameters in the network model. By conducting experiments on different parameters of the model, the parameter values that maximize both network stability and provide the best generalization ability to the network were selected;Model training: ResNet, SE_ResNet, and DenseNet were constructed for evaluating the hydraulic fracturing effect of the tight reservoir layer, and the three models were trained using the processed dataset and WGAN-GP-generated samples;Hydraulic fracturing effect evaluation: The trained models were used to evaluate the hydraulic fracturing effect of different wells and obtain the hydraulic fracturing section and non-fracturing section.

**Figure 8 sensors-24-05775-f008:**
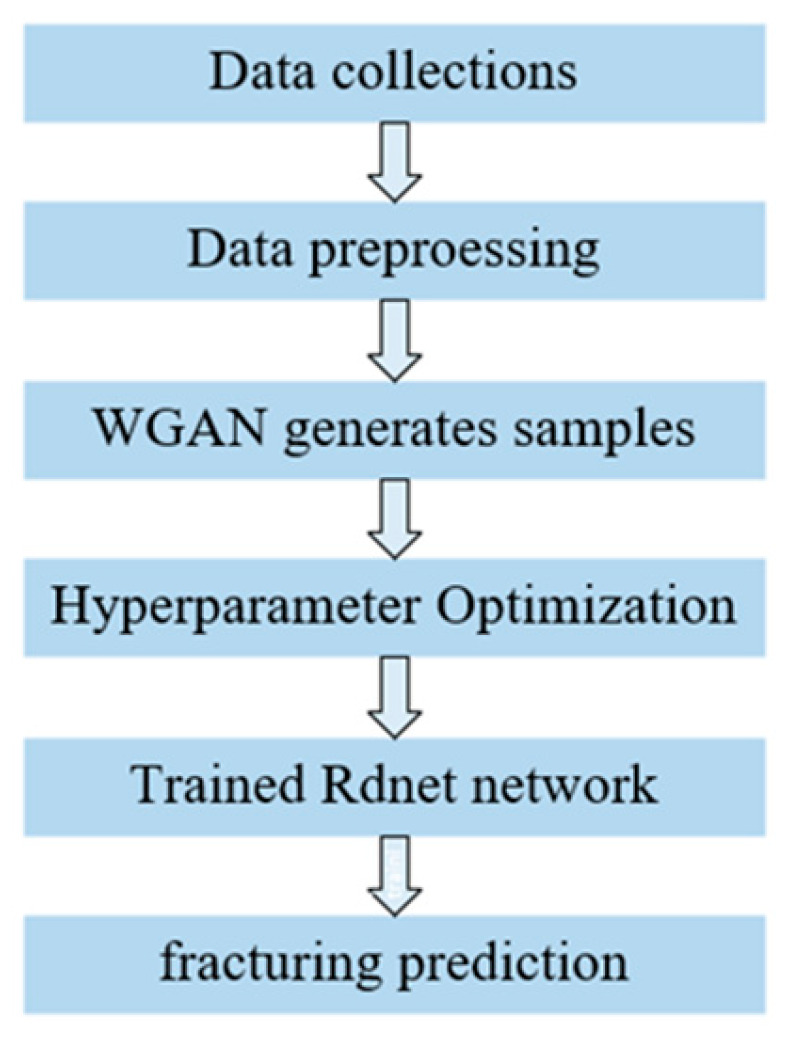
Workflow.

## 4. Experimental Process

To ensure the performance of the model, a reasonable parameter setting was required, which used the process shown in Figure 9. Firstly, data preprocessing was necessary, which included trimming, filtering, and normalization of the collected array acoustic logging data. Secondly, the training environment for this study was defined. Thirdly, a dataset was created for model training, and the dataset labels were manually annotated. Fourthly, the optimizer, learning rate, and loss function used in the deep learning training process were determined to improve the model’s performance. Finally, the signals generated by the improved WGAN-GP are presented as adversarial samples to demonstrate the effectiveness of improving the model’s generalization ability [31].

### 4.1. Experiment Environment

The platform configurations used for all training in this article are shown in Table 1. The training was performed on Windows 11 operating system with an Intel Core i5-12450 CPU @ 2.0 GHz and NVIDIA GeForce RTX 3050. The software environment used was implemented in Pytorch 1.12.1 framework under Python 3.9, and CUDA 11.6 was used for GPU accelerated computing.

### 4.2. Evaluation Indicators

The problem investigated in this paper is essentially a classification task. Therefore, the performance of each model was evaluated using a confusion matrix, which provided an effective assessment of the classification model’s performance. The confusion matrix is typically a 2x2 matrix, which includes four types of classification results: TP, FP, TN, and FN. Specifically, the confusion matrix is shown in Table 2.

The confusion matrix assisted in calculating evaluation metrics for the model. The specific formulas for calculation are as follows:(6)$$Accuracy=\frac{TP+TN}{TP+FN+FP+TN}$$
(7)$$Precision=\frac{TP}{TP+FP}$$
(8)$$Recall=\frac{TP}{TP+FN}$$
(9)$$F1_{score}=\frac{2Precision\times Recall}{Precision+Recall}$$

The higher the accuracy, the more accurate the model’s predictions; the higher the recall, the stronger the model’s ability to identify positive samples. The higher the precision, the higher the accuracy of the model’s identification of positive samples. The higher the F1-score, the better the overall performance of the model.

### 4.3. Create the Dataset

In practical logging operations, the acoustic probe may collide, or the logging instrument may tilt in the wellbore, resulting in noise and other interference in the acoustic signal that is unrelated to the evaluation of fracturing effects [32]. Therefore, it is necessary to filter the array acoustic logging signal to remove unrelated interference and improve the performance of the model. The logging instrument used in this study operates at a frequency range of 2–6 kHz, so bandpass filtering was applied to the data after truncation. Bandpass filtering is a signal processor that can pass signals within a specific frequency range through a filter and weaken or filter out signals in other frequency ranges. One of the filtered samples is shown in Figure 10.

Figure 10 is divided into two parts. The upper part is the signal after fracturing, and the lower part is the signal before fracturing. Since there are eight receivers receiving the signal, each part consists of eight segments, and each segment has a length of 400. To fully utilize the signals collected by the eight receivers, the data dimension input to the deep learning model in this study is (8, 400, 2), where 8 represents 8 channels (i.e., receivers), 400 represents the signal length, and 2 represents the array acoustic logging signals before and after fracturing. In addition, since the amplitude of array acoustic logging signals obtained by different logging instruments varied greatly, the data needed to be normalized to speed up the convergence of the model. After processing, there were approximately 1000 data points in the dataset. The labels of the dataset were manually annotated based on the results of processing and interpretation, such as the time difference between pre- fracturing and post-fracturing acoustic waves [33], orthogonal dipole anisotropy [34], radial tomography imaging [35], and scattering attenuation [36].

### 4.4. Experimental Setup

#### 4.4.1. Experimental Pretreatment Process

Deep learning models have a large number of parameters, and even small variations in key parameters can significantly impact the model. Optimizing model performance and determining the optimal parameters are key applications of sensitivity analysis [37,38]. In this study, we used ResNet as a representative model to evaluate the importance of different optimizers, learning rates, and loss functions using the original dataset. We found that this training strategy yielded the best performance across all networks, including SE_ResNet and DenseNet. Therefore, we only present the parameter selection process for ResNet.

Optimizer selection: The optimizer is a crucial part of training a deep learning model, as it determines how to update the model parameters to minimize the loss function. Different optimizers can have varying effects on the performance of the model.

In this study, with a fixed learning rate of 0.001, the performance of the model was compared using different optimization algorithms (Adam, RMSProp, SGD, and Adagrad). Figure 11a shows the accuracy curve of the model on the validation set, and Figure 11b shows the loss curve of the model on the validation set. From the figures, it can be seen that the Adam optimizer has a faster convergence speed, higher prediction accuracy, and lower loss compared to the other optimizers. This is because the Adam optimizer has the characteristics of adaptive learning rate and momentum, which can calculate the adaptive learning rate based on the historical square sum of each parameter’s gradient, making it more suitable for the current gradient situation and thereby accelerating convergence. Additionally, the Adam optimizer performs better in handling sparse gradients compared to Adagrad and RMSProp optimizers. Therefore, in the subsequent model training, Adam was chosen as the optimizer for the model training.

2.Adjusting learning rate: Learning rate is a critical hyperparameter in deep learning model training, as it determines the extent to which the model parameters are adjusted with each update. The choice of learning rate has a significant impact on the training process and performance of deep learning models. If the learning rate is too high, it may cause the model to not converge or produce unstable results, failing to reach the global optimal solution. Conversely, if the learning rate is too low, the model may require more time to reach its optimal performance or get stuck in a local minimum. After selecting Adam as the model optimizer, in order to determine the impact of hyperparameter optimization on the model, the learning rate was adjusted to 0.01, 0.001, and 0.0001. Figure 12a,b, respectively, display the accuracy and loss values of the model under different learning rates. The training steps are represented on the horizontal axis, while accuracy and loss values are on the vertical axis.

The experimental results indicate that when the learning rate is set to 0.001, the model converges faster, achieves higher accuracy, and exhibits more stable loss reduction. This suggests that the model can quickly learn the dataset’s features and optimize the objective function better with a learning rate of 0.001. However, when the learning rate is set to 0.01, the loss curve shows significant fluctuations after the tenth step. This is because the learning rate is too high, causing the model to skip the optimal solution and get trapped in a local minimum or oscillate, resulting in unstable training. On the other hand, when the learning rate is set to 0.0001, the loss reduction is slow, and the loss value remains higher compared to the other two learning rates. This is due to the learning rate being too low, making it challenging for the model to quickly adapt to the dataset’s features and struggle to achieve optimal performance, getting stuck in a local minimum. Therefore, in this experiment, a learning rate of 0.001 appears to be the most suitable choice. 

3.This paper uses array acoustic logging data, where there is a significant difference in the number of positive and negative samples. To address the class imbalance issue and compare the performance of different loss functions, the paper compares focal loss with cross-entropy loss. The experimental results are shown in Figure 13a,b, with the horizontal axis representing the training steps and the vertical axis representing accuracy and loss values, respectively. From the figures, it can be observed that the model using focal loss achieved higher accuracy and showed a significant decrease in the loss value compared to using the cross-entropy loss function. This indicates that the cross-entropy loss function is prone to be influenced by the majority class, making it difficult for the model to correctly classify minority class samples. In contrast, focal loss reduced the loss value for easily classifiable samples and increased the loss value for difficult-to-classify samples, effectively addressing the class imbalance problem and enabling the model to focus more on challenging samples. Therefore, using focal loss as the loss function leads to improved performance.

#### 4.4.2. WGAN-GP-Generated Signals

The aim of this section is to study the distribution characteristics and amplitude–frequency characteristics of the processed signal dataset in order to train a WGAN-GP network for generating effective array acoustic logging data. The noise dimension used for generator training in this paper is (25,1,1), and the input data dimension used for discriminator training is (400,8,2). The training parameters were set as follows: The batch size is 32, the number of training steps (epochs) was set to 1000, the learning rate for both the generator and discriminator was set to 0.00018, and the optimizer used is Adam with default parameters. The gradient penalty strength λ in WGAN-GP was set to 10. Additionally, during the training of WGAN-GP, this study adopted a strategy of optimizing the generator once after training the discriminator five times. This strategy prevents gradient vanishing due to the discriminator’s high accuracy and leads to a more stable training process for WGAN-GP. The training process of WGAN-GP is shown in Figure 14.

During the training process, the chart of the WGAN-GP network shows the number of training steps (epochs) on the horizontal axis and the loss values on the vertical axis. At the beginning of training, the discriminator loss was negative. As the training progressed, the discriminator loss increased gradually and stabilized at a relatively large positive value. It then started to decrease and converged at around step 600. Simultaneously, the generator gradually generated more realistic data, leading to a gradual approach of the discriminator loss towards zero.

In order to visually demonstrate the effectiveness of the WGAN-GP generation network, this paper visualizes the signals generated at different stages of training. Figure 15 shows four example signals generated at 5, 200, 500, and 1000 epochs.

In this paper, the generator was trained using WGAN-GP for 1000 iterations, and 1000 adversarial samples were generated, consisting of 500 signals from fractured zones and 500 signals from unfractured zones. To verify the effectiveness of the WGAN-GP model in generating signals, the amplitude–frequency characteristics of the generated signals were compared with the spectrogram of real signals, as shown in Figure 16a,b. The results indicate that the amplitude–frequency characteristics of the generated signals are essentially consistent with those of real signals, demonstrating that the generator learned the distribution of real signals. However, there were still some noises outside the main frequency range in the spectrogram of the generated signals. Therefore, post-generation filtering was applied to the generated signals to accelerate the model’s convergence. The final experimental results show that the improved WGAN-GP, which includes a regularization term for gradient penalty, effectively addressed issues in the generated sample set and enhanced the quality of the generated signals.

#### 4.4.3. Dataset Enhanced

Through experiments, Ilyas et al. [39] demonstrated that non-robust features can enhance a network’s generalization ability, leading to more accurate predictions. In contrast, the original model showed poor generalization ability, resulting in lower prediction accuracy. Therefore, in this study, the improved WGAN-GP was used to generate an adversarial sample set based on the original dataset to enhance the dataset. The generated data were used as a set of adversarial samples to improve the network model’s generalization ability during training. The effectiveness of data enhancement was evaluated using accuracy and loss values. The model training parameters were set as follows: learning rate of 0.001, 200 training steps, and Adam optimizer, with cross-entropy loss function chosen as the loss function.

Figure 17a presents the accuracy curves of the model before and after using the adversarial sample set, while Figure 17b shows the corresponding loss curves. The blue curves represent the model’s accuracy and loss when trained solely on the original dataset, while the red curves represent the accuracy and loss after incorporating the adversarial sample set. The experimental results indicated that using the adversarial sample set leads to significant improvements in accuracy, a more stable loss curve, faster convergence, and ultimately lower loss values after convergence. Consequently, the performance of the model was enhanced through the application of the adversarial sample set, and this data augmentation method will be utilized in subsequent studies to achieve a more robust evaluation of fractured zones.

## 5. Analysis of Experimental Results

The deep learning-based evaluation technology for fracture height in tight reservoirs mainly relies on classification models. Therefore, selecting the appropriate deep learning models is crucial for the entire evaluation process. In this section, we evaluate the performance of ResNet, DenseNet, and ResNet with SE module, which are currently the most representative deep learning classification models, for the evaluation of fracture height in tight reservoirs. The model performance was analyzed comprehensively based on accuracy, recall, precision, and F1-Score to determine the most suitable deep learning classification model for fracture height evaluation in tight reservoirs.

This study utilized a dataset of 2000 samples for training the model, comprising 1000 processed array acoustic logging signals and another 1000 adversarial samples generated using WGAN-GP. The training-to-testing dataset ratio was set to 4:1. All network comparisons were conducted without using pre-trained weights to ensure that the models fully adapted to the specific characteristics of the dataset. During training, we conducted 100 iterations with an initial learning rate of 0.001, using Adam optimizer and focal loss as the loss function. After training, the model’s performance was evaluated on array acoustic logging data from three additional well sections, and the results are summarized in Table 3.

As shown in Table 3, on the dataset of three wells, the model with dense connections had the lowest accuracy, reaching as low as 84.96%, while the model with residual connections had the highest accuracy in the evaluation of hydraulic fracturing height, with the lowest accuracy of 91.6% on the three-well dataset and a maximum of 99.75%. This indicates that residual connections have better performance than dense connections on the array sonic logging dataset, and the calculation order of convolution, BN, and activation functions in ResNet is more suitable for the data characteristics than the calculation order of BN, activation functions, and convolution in DenseNet. In addition, the residual network with SE module had more accurate evaluation results, and its evaluation results on the dataset of three wells were better than other models. Relatively speaking, the residual network with SE module can adaptively learn the importance of each channel; weight the characteristics of the eight channels of input data, thereby improving the feature representation ability of the network; and adaptively learn the importance of features during training, thus improving the model’s generalization ability and reducing the risk of overfitting.

Figure 18 shows the visualization results of the model evaluation on the three wells.

In Figure 18, each subfigure contains evaluation results of three models. The blue curve represents the evaluation results of the models, while the red dotted line represents the comprehensive interpretation result of the well-logging data. The horizontal axis represents the fracturing effect, where 1 corresponds to the depth of the fracturing zone, and 0 corresponds to the depth of the non-fracturing zone; the vertical axis represents the depth of the well. The evaluation results of the SE_ResNet model were more concentrated on the fracturing effect than the other two models, which is consistent with the actual fracturing area in the application. Although the ResNet model can correctly evaluate the fracturing effect, it may also produce some erroneous evaluations, such as misjudging the non-fracturing area as the fracturing area. The evaluation results of the DenseNet model were even worse, and the evaluation of the fracturing area was more scattered.

In addition, in terms of well characteristics, the evaluation effect of the three models on well 1 was better than that on the other two wells. No erroneous evaluation of other intervals was produced in well 1, while erroneous evaluation of a depth range occurred in the evaluation results of well 2, and the erroneous evaluation of well 3 was relatively more scattered. This is because the quality of the cementing in well 1 is better, and the morphology of the array acoustic logging data before and after fracturing was better. More importantly, the array acoustic logging data acquisition instruments in well 1 before and after fracturing were the same. However, the cementing quality in the other two wells is poor, and there may be the influence of casing waves in the array acoustic logging data, resulting in a lot of information that is irrelevant to the evaluation of fracturing effect. Additionally, the array acoustic logging data collected before and after fracturing have large differences in morphology, leading to many erroneous evaluations. Therefore, the quality of cementing and whether the data acquisition instruments before and after fracturing are the same are crucial for the performance of the model.

## 6. Experimental Analysis and Discussion

To address a series of issues in traditional fracturing effectiveness evaluation methods, this study employed an automated evaluation approach based on deep learning algorithms to assess the fracturing effectiveness of tight reservoirs. The main contributions of this article are as follows:This paper examines the current issues in the field of fracturing effectiveness evaluation in tight reservoirs and investigates the latest research trends in well logging related to fracturing effectiveness evaluation and deep learning, both domestically and internationally;This article provides a comprehensive introduction to the residual connections in ResNet, the dense connections in DenseNet, and the characteristics of the SE module. It elucidates the reasons behind the applicability of these models to array acoustic logging data. Subsequently, the utilization of WGAN-GP for expanding the dataset was described, achieving the goal of data augmentation;This study built a dataset by applying bandpass filtering and normalization techniques to achieve balanced data across various categories and created dataset labels and enhanced the dataset using WGAN-GP technology. Subsequently, we compared the generated signal spectrum with the actual signal spectrum to validate the effectiveness of the generated signals;Evaluation of the effectiveness of high-pressure fracturing was addressed in this study. Firstly, a robust evaluation model was established for assessing fracturing effects in tight reservoirs. Additionally, a generator and discriminator for the WGAN-GP technique were constructed and further enhanced. Through parameter experimentation, key choices such as optimizer selection, learning rate determination, and loss function definition were made. The selected model parameters consist of an Adam optimizer, a learning rate of 0.001, and the application of focal loss as the loss function.

Subsequently, the performance improvement introduced by WGAN-GP was demonstrated by comparing model outcomes before and after integrating the generated samples. Furthermore, the SE (Squeeze and Excitation) module was trained independently based on the expanded dataset. Comparative evaluations between ResNet, DenseNet, and a SE_ResNet variant were performed through experimental results, showcasing these models’ performance in terms of accuracy, recall, and F1-Score. The empirical findings highlighted ResNet’s superior model architecture over DenseNet. For instance, in data from three wells, DenseNet’s accuracy ranged between 84.96% and 98.35%, while ResNet exhibited a higher accuracy range of 91.41% to 99.31%. The SE module, by adaptively weighting each channel of array acoustic logging data, further elevated the model’s precision in evaluating fracturing effectiveness. Within the dataset of three wells, the SE_ResNet model achieved a minimum accuracy of 91.6% and a maximum accuracy of 99.75%, underlining its efficacy.

Despite the partial investigation carried out in this study, certain unresolved queries persist, alongside areas that have yet to be traversed. Consequently, future research endeavors can be strategically focused on the subsequent trajectories:Collecting more dipole array acoustic logging signals to enhance dataset diversity. Although this study employed WGAN-GP for sample expansion, ensuring superior model generalization primarily involves amassing dipole array acoustic logging signals from diverse reservoirs across various regions. It is imperative to maintain consistency in data collection instruments before and after fracturing, alongside selecting high-quality wells to ensure data authenticity and precision;Acquiring expertise in unsupervised deep learning methodologies. While the deep learning techniques employed in this study yielded certain outcomes, the utilization of supervised deep learning approaches demands an extensive corpus of annotated data, potentially incurring inefficiencies in terms of time and labor costs. Consequently, for future investigations, the exploration of unsupervised deep learning methodologies is recommended. These methodologies possess the capability to autonomously discern category attributes within unlabeled data, thereby mitigating the demand for annotated data and its associated costs;Due to variations in signal acquisition equipment and geological conditions, data across different operational zones exhibit diversity. Inconsistencies in signal acquisition instruments before and after fracturing engender diminished accuracy in the model’s assessment of high-pressure fracturing [40]. Retraining the network for each operational zone presents the challenges of extensive data collection annotation and escalated network training complexity, significantly impeding algorithmic efficiency. Employing transfer learning centered on task similarity enables seamless model transfer between akin tasks. Leveraging the foundation of pre-trained network parameter initialization, the model’s adaptability to the target area’s distinctive problematics significantly curtails training data volume and network training expenditure [41].

## Figures and Tables

**Figure 1 sensors-24-05775-f001:**
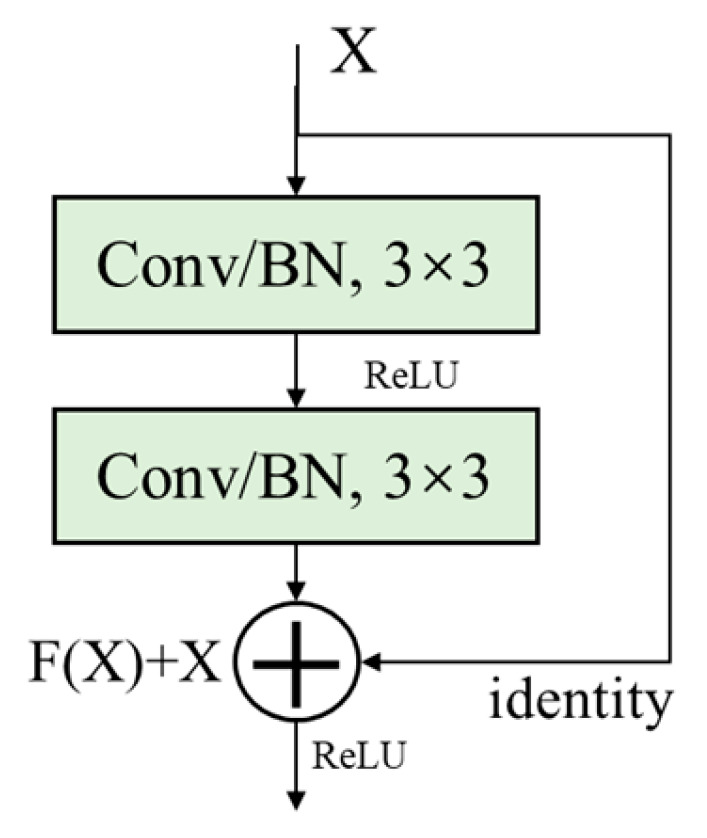
Structure of the residual block.

**Figure 2 sensors-24-05775-f002:**
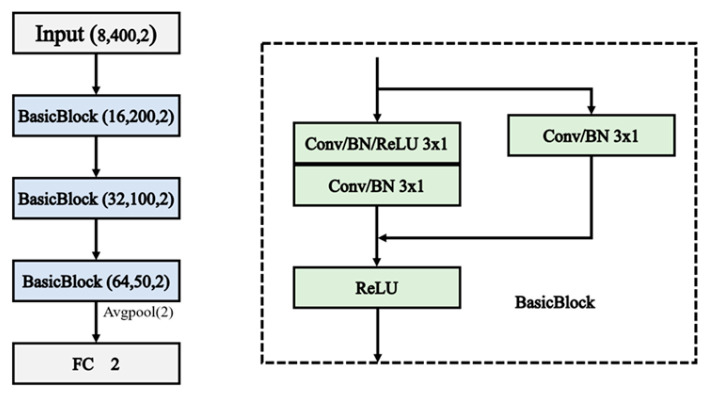
Structure of the ResNet.

**Figure 3 sensors-24-05775-f003:**
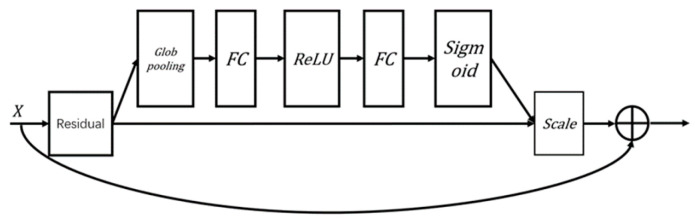
Structure of SE_BasicBlock.

**Figure 4 sensors-24-05775-f004:**
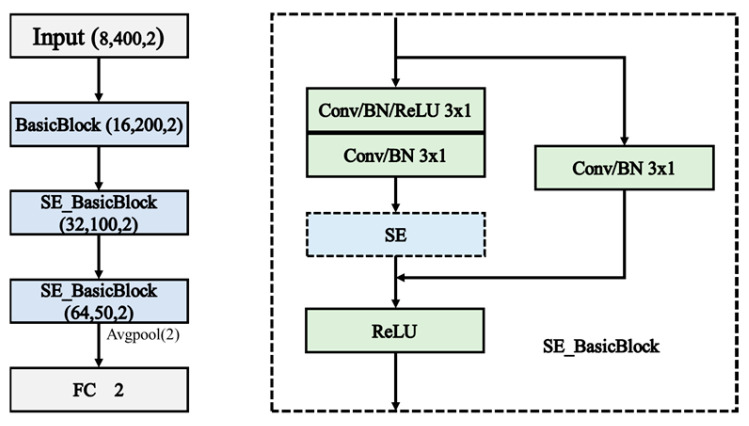
Structure of the SE_ResNet.

**Figure 5 sensors-24-05775-f005:**
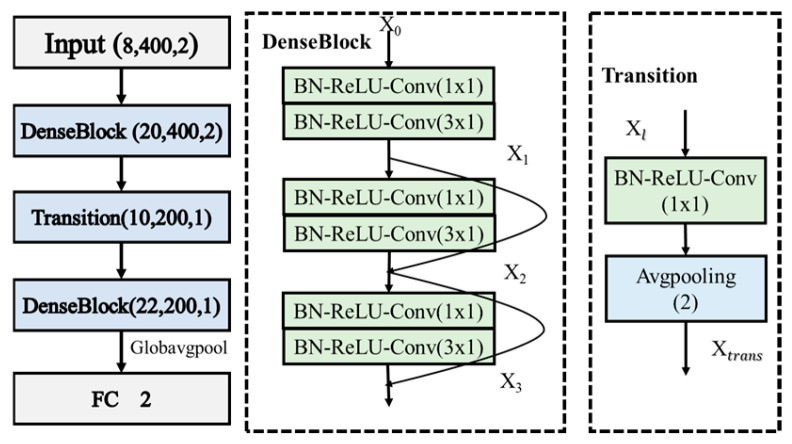
Structure of the DenseNet.

**Figure 6 sensors-24-05775-f006:**
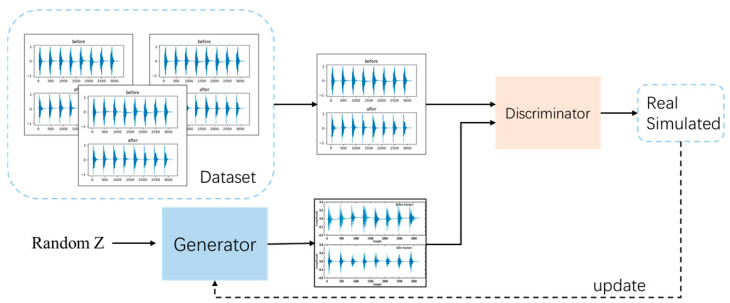
The overall structure diagram of WGAN-GP.

**Figure 7 sensors-24-05775-f007:**
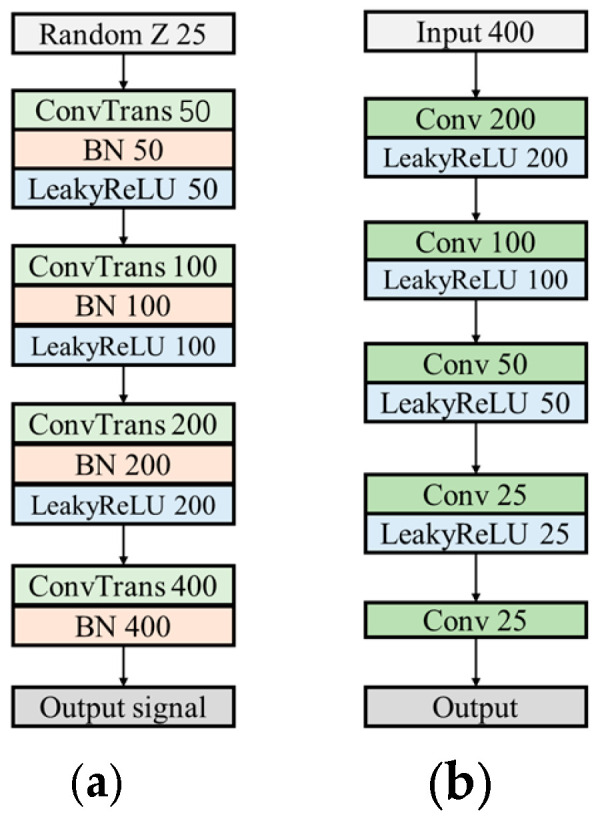
Detailed Substructures of WGAN-GP: (**a**) generator and (**b**) discriminator.

**Figure 9 sensors-24-05775-f009:**
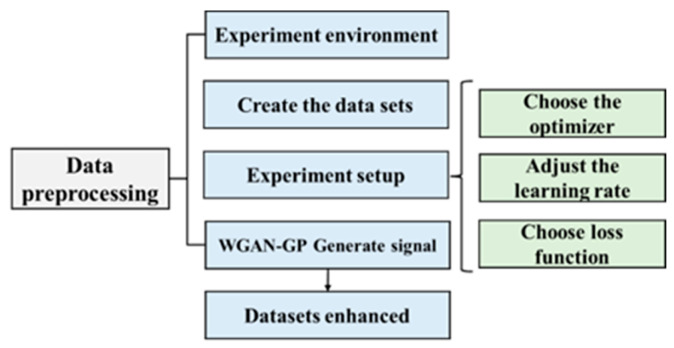
Experimental pretreatment process.

**Figure 10 sensors-24-05775-f010:**
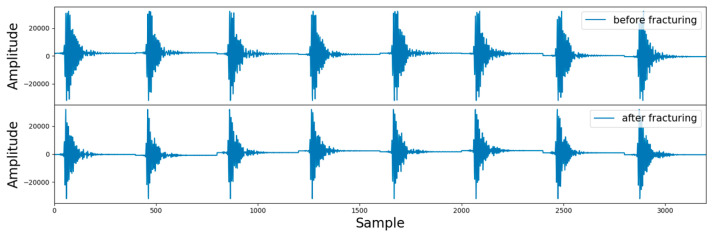
Schematic diagram of a single sample.

**Figure 11 sensors-24-05775-f011:**
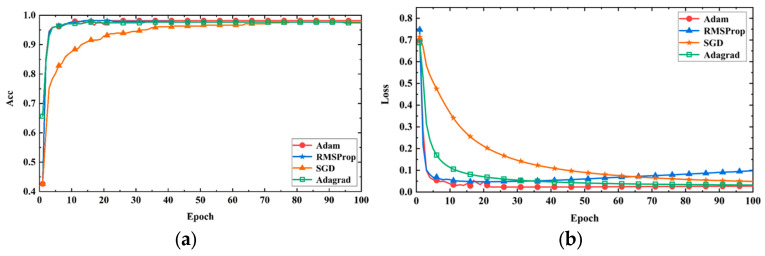
Model performance with different optimizers. (**a**) Accuracy curves; (**b**) loss curves.

**Figure 12 sensors-24-05775-f012:**
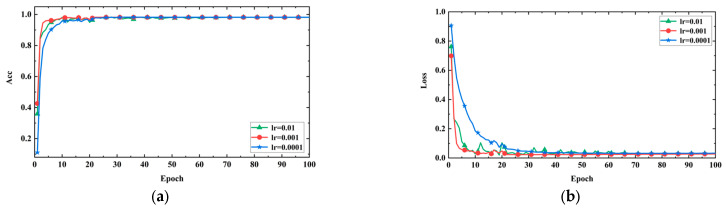
Model performance with different learning rates. (**a**) Accuracy curves; (**b**) loss curves.

**Figure 13 sensors-24-05775-f013:**
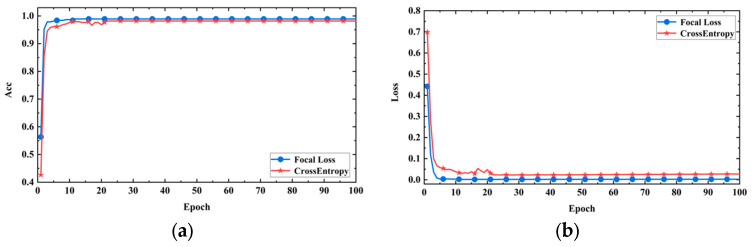
Model performance with different loss functions. (**a**)Accuracy curves; (**b**) loss curves.

**Figure 14 sensors-24-05775-f014:**
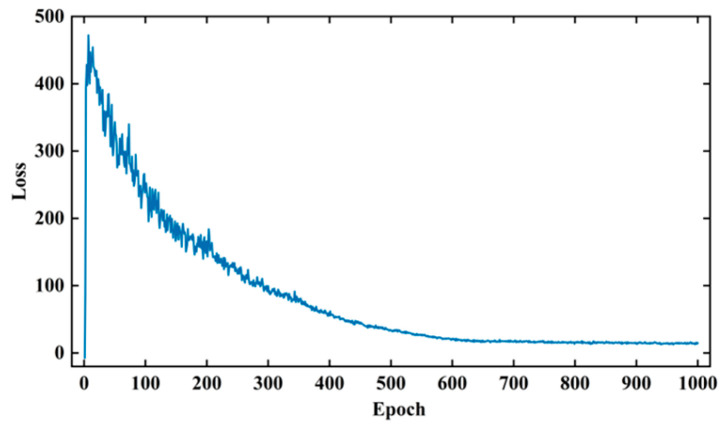
WGAN-GP discriminator loss curve.

**Figure 15 sensors-24-05775-f015:**
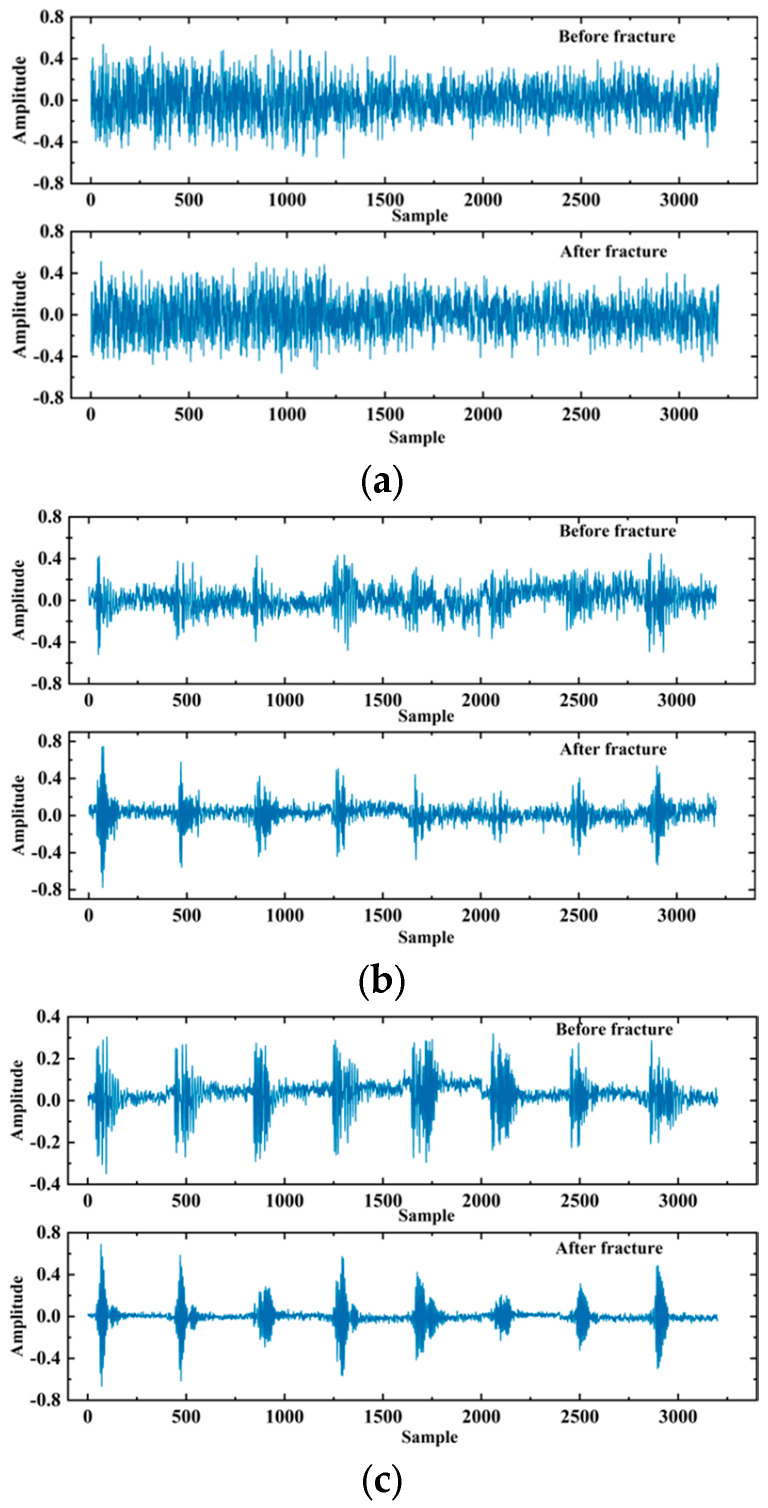
The generator generates the signal: (**a**) 5th iteration; (**b**) 200th iteration; (**c**) 500th iteration; (**d**) 1000th iteration.

**Figure 16 sensors-24-05775-f016:**
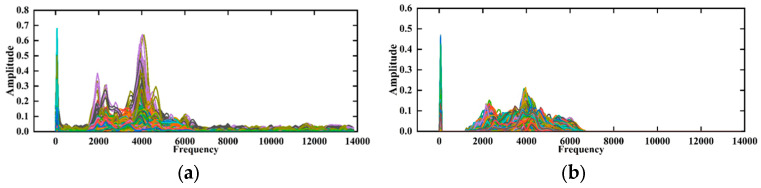
Generated signal and real signal spectrograms. (**a**) Generated signal spectrogram; (**b**) real signal spectrogram.

**Figure 17 sensors-24-05775-f017:**
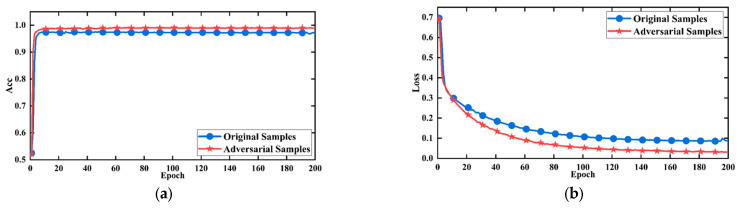
The performance of the model after using the adversarial set. (**a**) Accuracy curve; (**b**) loss curves.

**Figure 18 sensors-24-05775-f018:**
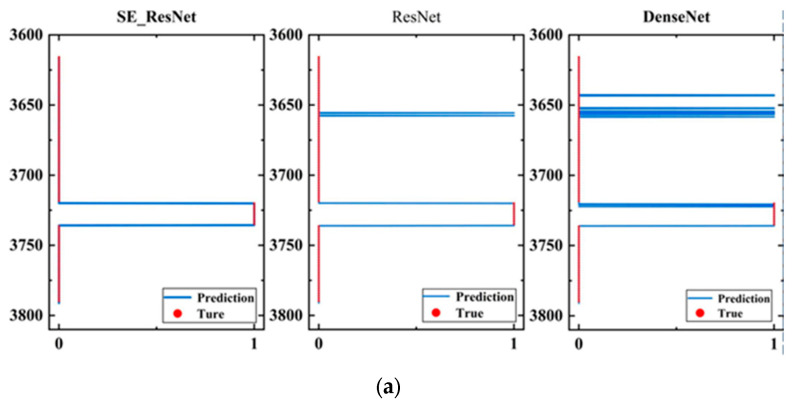
High evaluation results of fractures. (**a**) Fracture height prediction results for well 1; (**b**) fracture height prediction results for well 2; (**c**) fracture height prediction results for well 3.

**Table 1 sensors-24-05775-t001:** Experimental platform configuration.

Hardware	CPU	IntelCorei5-12450H
	GPU	GeForce RTX 3050
	RAM	16 GB
Software	Operating system	Windows 11
	Deep learning	Pytorch 1.12.1
	Programming	Pycharm + Python 3.9

**Table 2 sensors-24-05775-t002:** Confusion Matrix.

	Predicting Positive Samples	Predict Negative Samples
**True-positive sample**	TP (True Positive)	FN (False Negative)
**True-negative sample**	FP (False Positive)	TN (True Negative)

**Table 3 sensors-24-05775-t003:** High evaluation results of fractures in different models.

Well	Model	Accuracy	Recall	Precision	F1-Score
1	SE_ResNet	0.9975	0.9974	0.9975	0.9974
	ResNet	0.9931	0.9931	0.9935	0.9932
	DenseNet	0.9835	0.9835	0.9844	0.9838
2	SE_ResNet	0.9160	0.9160	0.9428	0.9217
	ResNet	0.9141	0.9141	0.9419	0.9200
	DenseNet	0.8789	0.9256	0.8789	0.8891
3	SE_ResNet	0.9645	0.9645	0.9759	0.9677
	ResNet	0.9318	0.9318	0.9646	0.9415
	DenseNet	0.8496	0.8496	0.9423	0.8804

## Data Availability

The data that support the findings of this study are available from the corresponding author (heichuang@126.com), upon reasonable request.

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
