# Peer review of "Evaluation of Fracturing Effect of Tight Reservoirs Based on Deep Learning"

_sensors, 2024, doi:10.3390/s24175775_

Round 1

Reviewer 1 Report

Comments and Suggestions for Authors

The paper entitled " Evaluation of fracturing effect of tight reservoirs based on deep learning" is within the scope of this journal. The authors utilized a deep learning-based method to address the issue of evaluating fracturing effects in tight reservoirs. The experimental results show that the ResNet with residual connections is more suitable for the dataset in this paper and achieves higher accuracy in fracturing effect evaluation. The topic of the research is interesting. However, the current draft requires addressing some moderate issues listed here.

1.       The abstract needs to quantitatively specify the accuracy advantages of the residual block and SE module.

2.       The text in Figures 1, 6, 7, 8, and 10 is vertically aligned, making it difficult to read.

3.       The annotations in Figures 1, 2, 3, 4, 5, 6, 7, and 8 are labeled as “~ structure”; please check for grammatical errors.

4.       Figures 3 and 4 are redundant; it is recommended to merge and simplify them.

5.       The annotation for Figure 4 is “Basic structure of SE_ResNet,” which seems incorrect. It looks more like the structure of “SE_BasicBlock.”

6.       In Figure 7, “Generate” should be “Generator.”

7.       The overall structure diagram of WGAN-GP is missing; it is suggested to incorporate both the generator and discriminator into the overall structure diagram.

8.       The annotation for Figure 10 as “Experimental process” is not appropriate; a more suitable annotation or layout method is recommended.

9.       In section 4.4, the learning rate is fixed at 0.001 to compare the performance of Adam, RMSProp, SGD, and Adagrad optimizers. Then, Adam is used to compare different learning rates. Generally, for the same dataset, the learning rates suitable for Adam and SGD are not the same, with SGD likely needing a slightly higher rate. The paper does not use various combinations of learning rates and optimizers, making the final combination (Adam + lr 0.001) unconvincing.

10.    Learning rate settings and data augmentation are placed in the same section, but the learning rate mentioned earlier is not used in the WGAN-GP training. Instead, ResNet, DenseNet, and SE_ResNet all use Adam + lr 0.001. What exactly is section 4.4 trying to illustrate? The logical flow between paragraphs should be reconsidered.

11.    The depths of the ResNet, DenseNet, and SE_ResNet networks are different, and they have varying numbers of parameters, yet the same training strategy is used. Has it been verified that these models are optimal under this training strategy?

12.    All figures need to be resized and drawn more compactly.

13.    For improvement of the relevant literature about the fracture identification, I suggest consulting the following papers.

Li T, Wang R, Wang Z, Zhao M, Li L. Prediction of fracture density using genetic algorithm support vector machine based on acoustic logging data. Geophysics. 2018 Mar 1;83(2):D49-60.

Li T, Wang Z, Wang R, Yu N. Pore type identification in carbonate rocks using convolutional neural network based on acoustic logging data. Neural Computing and Applications. 2021 May;33(9):4151-63.

Comments on the Quality of English Language

Moderate editing of English language required

Author Response

  1. The abstract needs to quantitatively specify the accuracy advantages of the residual block and SE module.

Author Response:

Thank you for your valuable feedback on our paper. We have revised the abstract as per your suggestion, specifically including a quantitative evaluation of the accuracy achieved by the SE module. The revised abstract now highlights that the highest accuracy of 99.75% was obtained, emphasizing the significance of the SE module in improving the model's accuracy.

“The utilization of hydraulic fracturing technology is indispensable for unlocking the potential of tight oil and gas reservoirs. Understanding and accurately evaluating the impact of fracturing is pivotal in maximizing oil and gas production and optimizing wellbore performance. Currently, evaluation methods based on acoustic logging, such as orthogonal dipole anisotropy and radial tomography imaging, are widely used. However, when the fractures generated by hydraulic fracturing form a network-like pattern, orthogonal dipole anisotropy fails to accurately assess the fracturing effects. Radial tomography imaging can address this issue, but it is challenged by high manpower and time costs. This study aims to develop a more efficient and accurate method for evaluating fracturing effects in tight reservoirs using deep learning techniques. Specifically, the method utilizes dipole array acoustic logging curves recorded before and after fracturing. Manual labeling was conducted by integrating logging data interpretation results. An improved WGAN-GP is employed to generate adversarial samples for data augmentation, and fracturing effect evaluation is implemented using SE-ResNet, ResNet, and DenseNet. The experimental results demonstrate that ResNet with residual connections is more suitable for the dataset in this study, achieving higher accuracy in fracturing effect evaluation. The inclusion of the SE module further enhances model accuracy by adaptively adjusting the weights of feature map channels, with the highest accuracy reaching 99.75%.”

  1. The text in Figures 1, 6, 7, 8, and 10 is vertically aligned, making it difficult to read.

Author Response:

Thank you for your detailed feedback on the text alignment issues in Figures 1, 6, 7, 8, and 10. We apologize for the readability problems caused by the vertical alignment of the text. To address this issue, we have adjusted the text in the aforementioned figures to ensure it is horizontally aligned, thereby improving readability. Additionally, due to the removal of Figure 3, the subsequent figure labels have been renumbered accordingly.

  1. The annotations in Figures 1, 2, 3, 4, 5, 6, 7, and 8 are labeled as “~ structure”; please check for grammatical errors.

Author Response:

Thank you for pointing out the issue with the annotations in Figures 1, 2, 3, 4, 5, 6, 7, and 8 being labeled as “~ structure.” We have reviewed your feedback and made the necessary corrections.

We have revised the labels in these figures to use more specific and clear terminology. The updated labels now accurately reflect the content of each figure and enhance the overall clarity of the manuscript.

Figure 1: The label has been changed from "BasicBlock structure" to "Structure of the Residual Block.".

Figure 2: The label has been changed from "ResNet structure diagram" to " Structure of the ResNet ".

Figure 3: The label has been changed from "Basic structure of SE_ResNet " to " Structure of SE_BasicBlock ".

Figure 4: The label has been changed from " SE_ResNet structure diagram " to " Structure of the SE_ResNet ".

Figure 5: The label has been changed from " Dense block structure" to " Structure of the DenseNet.".

Figure 7: The label has been changed from " Generator structure " to " Detailed Substructures of WGAN-GP ", as it now combines content from the previous Figures 7 and 8.

Additionally, we have removed Figure 3, added a new Figure 6 to illustrate the overall structure of the WGAN network, and merged Figures 7 and 8 into a single Figure 7. The subsequent figure labels have been renumbered accordingly. We believe these changes will improve the accuracy of the figures.

  1. Figures 3 and 4 are redundant; it is recommended to merge and simplify them.

Author Response:

Thank you for your feedback regarding Figures 3 and 4. We appreciate your suggestion to merge and simplify these figures.

In response to your recommendation, we have combined Figures 3 and 4 into a single, simplified figure. This revised figure now consolidates the content and presents it in a more streamlined manner, enhancing clarity and reducing redundancy.

“The SE module is a module used to enhance the expressive power of deep neural networks. It can adaptively adjust the channel weights of feature maps, and more accurately capture critical feature information on the basis of deep networks[25]. As an attention mechanism, the SE module, due to its flexibility, can be embedded into various types of neural networks and is typically used in convolutional neural networks and residual networks. It mainly consists of two operations: Squeeze operation and Excitation operation. The Squeeze operation reduces the dimension of the feature map by using global average pooling, compressing the information of each channel into a single value. The Excitation operation, similar to the gating units in recurrent neural networks (RNNs), learns the importance weight of each channel through two fully connected layers, weights and sums the values of each channel to obtain a new feature map.

In SE_ResNet, the SE module is incorporated after the residual block structure, as illustrated in Figure 3. The "Residual" in the figure refers to the two convolutional layers before the addition of the feature maps in the residual block. The SE module's role here is to further enhance the network's ability to focus on the most relevant features, thereby improving overall model performance.

Figure.3. Structure of SE_BasicBlock“

  1. The annotation for Figure 4 is “Basic structure of SE_ResNet,” which seems incorrect. It looks more like the structure of “SE_BasicBlock.”

Author Response:

Thank you for pointing out the issue with the annotation for Figure 4. We apologize for the oversight.

“Basic structure of SE_ResNet” does not accurately describe the content of the figure, which actually represents the structure of “SE_BasicBlock.” We have corrected this in the revised manuscript.

Figure.3. Structure of SE_BasicBlock”

  1. In Figure 7, “Generate” should be “Generator.”

Author Response:

Thank you for your meticulous review and for highlighting the issue with the annotation in Figure 7. We sincerely apologize for the error.

The term “Generate” should be replaced with “Generator” to accurately reflect the intended meaning. We have promptly corrected this in the revised manuscript to ensure that terminology is consistent and precise throughout.

  1. The overall structure diagram of WGAN-GP is missing; it is suggested to incorporate both the generator and discriminator into the overall structure diagram.

Author Response:

Thank you for your valuable feedback regarding the overall structure diagram of WGAN-GP.

In response to your recommendation, we have added a comprehensive structure diagram of WGAN-GP to the revised manuscript. This new diagram now integrates both the generator and discriminator, providing a complete overview of the WGAN-GP architecture.

Figure 7. The overall structure diagram of WGAN-GP

  1. The annotation for Figure 10 as “Experimental process” is not appropriate; a more suitable annotation or layout method is recommended.

Author Response:

Thank you for your insightful feedback regarding the annotation for Figure 10.

We acknowledge that the label "Experimental process" may not have fully captured the intended content of the figure. In response to your recommendation, we have revised the annotation to "Experimental pretreatment process" to better reflect the figure’s purpose and provide a clearer description.

Figure 9. Experimental pretreatment process

  1. In section 4.4, the learning rate is fixed at 0.001 to compare the performance of Adam, RMSProp, SGD, and Adagrad optimizers. Then, Adam is used to compare different learning rates. Generally, for the same dataset, the learning rates suitable for Adam and SGD are not the same, with SGD likely needing a slightly higher rate. The paper does not use various combinations of learning rates and optimizers, making the final combination (Adam + lr 0.001) unconvincing.

Author Response:

Thanks for the comment. In response to your suggestion, I conducted additional comparison experiments. Specifically, I tested the performance of the Adam optimizer with a 0.001 learning rate and compared it with the performance of SGD at learning rates of 0.1, 0.01, and 0.001. The detailed results are shown in the figure below. These results further confirm that the Adam + 0.001 combination outperforms the SGD optimizer for the dataset and task we studied. While it is true that different optimizers typically require different learning rates to achieve optimal performance, the experiment shows that Adam + 0.001 demonstrates greater robustness for the task of fracturing effectiveness evaluation.

(a)Accuracy curves;

(b)Loss curves

  1. Learning rate settings and data augmentation are placed in the same section, but the learning rate mentioned earlier is not used in the WGAN-GP training. Instead, ResNet, DenseNet, and SE_ResNet all use Adam + lr 0.001. What exactly is section 4.4 trying to illustrate? The logical flow between paragraphs should be reconsidered.

Author Response:

Thanks for the comment. We have reconsidered the paragraph structure and decided to merge Sections 4.4 and 4.5 into a single section.

As for the WGAN-GP training, the previously mentioned learning rate was not used because WGAN-GP is a type of Generative Adversarial Network (GAN) with the primary goal of generating realistic data distributions. The data generated by WGAN-GP is used to increase sample size, enrich data diversity, and address class imbalance issues, thereby improving the accuracy of the model used for fracturing evaluation. This is fundamentally different from the networks like ResNet, DenseNet, and SE_ResNet, which are used for fracturing evaluation tasks. GAN training is often more challenging and highly sensitive to the optimizer and learning rate settings. To ensure stable training, WGAN-GP requires different optimizer configurations and learning rates.

  1. The depths of the ResNet, DenseNet, and SE_ResNet networks are different, and they have varying numbers of parameters, yet the same training strategy is used. Has it been verified that these models are optimal under this training strategy?

Author Response:

Thank you for your insightful question regarding the training strategy used for ResNet, DenseNet, and SE_ResNet.

We would like to clarify that all network comparisons in our study were conducted without using pre trained weights, and each model was trained using an optimal training strategy. During the experimental setup, we validated that the selected training strategy was effective across all models. Given this, we did not include a detailed discussion of the training parameter selection process for each network in the manuscript, and instead, we only presented the experimental results for ResNet. However, to avoid any potential misunderstanding, we have revised the manuscript to emphasize this point more clearly.

In Section 4.4, we made the following revisions:
“Deep learning models have a large number of parameters, and even small variations in key parameters can significantly impact the model. Optimizing model performance and determining the optimal parameters are key applications of sensitivity analysis[37,38]. In this study, we used ResNet as a representative model to evaluate the importance of different optimizers, learning rates, and loss functions using the original dataset. We found that this training strategy yielded the best performance across all networks, including SE_ResNet and DenseNet. Therefore, we only presented the parameter selection process for ResNet.”

In Section 5, we made the following revisions:
“This study utilized a dataset of 2000 samples for training the model, comprising 1000 processed array acoustic logging signals and another 1000 adversarial samples generated using WGAN-GP. The training-to-testing dataset ratio was set to 4:1. All network comparisons were conducted without using pre-trained weights to ensure that the models fully adapted to the specific characteristics of the dataset. During training, we conducted 100 iterations with an initial learning rate of 0.001, using Adam optimizer and Focal Loss as the loss function. After training, the model's performance was evaluated on array acoustic logging data from three additional well sections, and the results are summarized in Table 3.”

  1. All figures need to be resized and drawn more compactly.

Author Response:

Thanks for the comment. We have resized all figures and adjusted their layout to make them more compact, ensuring they are more visually clear and better aligned with the text. Thank you for your valuable feedback.

  1. For improvement of the relevant literature about the fracture identification, I suggest consulting the following papers.

Author Response:

Thanks for the comment. We have carefully reviewed the literature and incorporated additional references on fracture identification into the introduction section of the revised manuscript, as suggested. These additional references will support and enhance the discussion of fracture identification in our work.

  1. Li, T.; Wang, R.; Wang, Z.; Zhao, M.; Li, L. Prediction of Fracture Density Using Genetic Algorithm Support Vector Machine Based on Acoustic Logging Data. GEOPHYSICS 2018, 83, D49–D60, doi:10.1190/geo2017-0229.1.
  2. Li, T.; Wang, Z.; Wang, R.; Yu, N. Pore Type Identification in Carbonate Rocks Using Convolutional Neural Network Based on Acoustic Logging Data. Neural Comput & Applic 2021, 33, 4151–4163, doi:10.1007/s00521-020-05246-2.

Reviewer 2 Report

Comments and Suggestions for Authors

It would be good to compare the fracturing characteristics obtained from the simulation with real fracturing data. The proposed methodology is interesting, but more extensive empirical validation is needed.

Author Response

It would be good to compare the fracturing characteristics obtained from the simulation with real fracturing data. The proposed methodology is interesting, but more extensive empirical validation is needed.

Author Response:

Thanks for the comment. The comparison is as follows:

Figure 1. Comparison of Simulated Fracturing Data. (a)Fracturing zone. (b)Not fracturing zone

Figure 1(a) shows the waveforms generated by the WGAN-GP simulation for the non-fractured region before and after fracturing, while Figure 1(b) displays the waveforms for the fractured region before and after fracturing. It can be observed that the waveforms in the non-fractured region exhibit a high degree of similarity before and after fracturing. In contrast, although the waveforms in the fractured region are similar in the direct wave segment, noticeable tailing appears in the post-fracturing waveforms as time progresses (highlighted in the green box).

Figure 2. Comparison of real fracturing data. (a)Fracturing zone. (b)Not fracturing zone

The acoustic logging data before and after fracturing in non-fractured areas are basically the same, and the waveform data after fracturing in fractured areas show obvious scattering waves after the direct wave, as shown in the green box, which can be found in the references:

Tang X M, Li Z, Hei C, et al. Elastic wave scattering to characterise heterogeneities in the borehole environment[J]. GEOPHYSICAL JOURNAL INTERNATIONAL, 2016, 205(1): 594-603.

Reviewer 3 Report

Comments and Suggestions for Authors

The paper is devoted to the improvement of models and machine learning algorithms for predicting the effect of hydraulic fracturing in tight reservoirs characterized by low permeability. Hydraulic fracturing technology increases oil and gas production when it meets certain reservoir characteristics, so it is important to predict the effectiveness of hydraulic fracturing in tight reservoirs based on their geological, physical and mechanical characteristics. The problem of predicting the efficiency of fracturing in tight reservoirs, despite a number of studies in this scientific field, remains relevant at present. This is because the problem is very complex, and traditional methods of prediction have certain limitations detailed in the manuscript (Lines 28-49).  

The use of machine learning has led to significant progress in predicting hydraulic fracturing performance. Nevertheless, based on the analysis of the current state of the designated problem, the authors rightly note that machine learning still faces difficulties in learning the basic rules of complex data and often has difficulties in achieving high prediction accuracy when dealing with complex data classification tasks (Lines 50-55). The logic of the study naturally led the authors to the necessity of applying deep learning methods to solve the problem of predicting fracturing performance.  

The main research question of this study is to develop an improved model for estimating the effects of hydraulic fracturing in tight reservoirs under high pressure. Since the relationship of hydraulic fracturing efficiency data of a particular reservoir is not entirely clear, deep learning algorithms are chosen to address the main research question in the manuscript because these algorithms are better suited to analyze patterns when the explicit relationship of the data is unclear.  

No ready solution to the above-mentioned main question could be found in the known literature (Lines 581-659). The manuscript contains a detailed description of the research methods (Sections 2-4), analysis of the simulation results (Section 5), and discussion (Section 6). Critical analysis of the methods for solving the main research question in the review part of the manuscript, detailing of the methodology, development of deep learning algorithms for a specific problem and verification of the modeling accuracy naturally led the authors to a result, which is important in scientific and practical terms. For instance, for data from three wells, the proposed model provides an accuracy of 91.6% to 99.75% (Table 3 and rows 534-539). The research methodology is applicable to other similar problems. This means that the manuscript closes a gap in this research area related to improving the prediction of fracturing performance of tight reservoirs based on deep learning.  

Commenting on the scientific contributions of the authors, the following should be noted.  In general, deep learning algorithms can extract different features, and combine these features to form higher-level concepts. However, the paper correctly notes that a network that is too deep can lead to problems with vanishing gradients or exploding gradients, which will lead to overfitting (Lines 64-72). The refereed manuscript provides a balanced solution to this problem in the form of a new robust model for predicting the fracturing effect of dense reservoirs based on deep learning. This model makes an important contribution to the subject area compared to other papers directly related to predicting the effect of fracturing based on deep learning.  

Despite the good results, the paper contains an analysis of unsolved problems and possibilities for their solution (lines 541-557), which increases the scientific significance of the study.  

The reviewer has no significant comments that could prevent publication of the manuscript. No additional controls are required for this study.  

Overall evaluation of the manuscript: The study presented in the manuscript meets the criteria of relevance of the topic, novelty of the results, scientific and practical significance. The methodology and results of the study are of interest to many potential readers interested in improving the efficiency of fracturing of tight reservoirs based on deep learning. The topic of the research corresponds to the Sensors (sensors are needed, for example, to obtain data according to Figure 10). The research methods correspond to the purpose of the study. The results and conclusions are consistent with the evidence and arguments presented. All structural elements of the article are logically connected to each other, and their content is oriented towards the purpose of the research. Figures 1-19. Appropriate, but see notes 1 and 2. Tables 1-3. Appropriate.  

Notes: 1. Lines 290-303, Figure 11: The text "Samples" in Figure 11 is inconsistent with the content of the figure, since Figure 11 shows signal segments of one sample received by eight receivers.  2. It is recommended to reformat Figures 1, 7, 8, 10, for example, Figure 6. 3. Formulate the purpose of the study more clearly in the abstract.

Author Response

  1. Lines 290-303, Figure 11: The text "Samples" in Figure 11 is inconsistent with the content of the figure, since Figure 11 shows signal segments of one sample received by eight receivers.

Author Response:

Thank you for pointing out the inconsistency with the text "Samples" in Figure 11. We have corrected this to "Sample" to avoid any ambiguity and ensure that the figure accurately reflects its content.

Figure 11. Schematic diagram of a single sample

  1. It is recommended to reformat Figures 1, 7, 8, 10, for example, Figure 6.

Author Response:

Thank you for your recommendation to reformat Figures 1, 7, 8, and 10. We have made the following changes:

The text alignment in all figures has been adjusted to enhance readability.

Figures 7 and 8 have been combined into a single figure to streamline the presentation.

Figure.1. Residual block structure.

Figure 5. Structure of the DenseNet.

a               b

Figure 7. Detailed Substructures of WGAN-GP: (a) Generator (b) Discriminator

Figure 9. Experimental pretreatment process

  1. Formulate the purpose of the study more clearly in the abstract.

Author Response:

Thank you for your suggestion to clarify the purpose of the study in the abstract. We have revised the abstract to explicitly state the study's purpose, ensuring that it is clearly communicated to the reader.

“The utilization of hydraulic fracturing technology is indispensable for unlocking the potential of tight oil and gas reservoirs. Understanding and accurately evaluating the impact of fracturing is pivotal in maximizing oil and gas production and optimizing wellbore performance. Currently, evaluation methods based on acoustic logging, such as orthogonal dipole anisotropy and radial tomography imaging, are widely used. However, when the fractures generated by hydraulic fracturing form a network-like pattern, orthogonal dipole anisotropy fails to accurately assess the fracturing effects. Radial tomography imaging can address this issue, but it is challenged by high manpower and time costs. This study aims to develop a more efficient and accurate method for evaluating fracturing effects in tight reservoirs using deep learning techniques. Specifically, the method utilizes dipole array acoustic logging curves recorded before and after fracturing. Manual labeling was conducted by integrating logging data interpretation results. An improved WGAN-GP is employed to generate adversarial samples for data augmentation, and fracturing effect evaluation is implemented using SE-ResNet, ResNet, and DenseNet. The experimental results demonstrate that ResNet with residual connections is more suitable for the dataset in this study, achieving higher accuracy in fracturing effect evaluation. The inclusion of the SE module further enhances model accuracy by adaptively adjusting the weights of feature map channels, with the highest accuracy reaching 99.75%.”

Round 2

Reviewer 1 Report

Comments and Suggestions for Authors

I believe the authors have done what could be done to meet both reviewers comments and recommendations.

They have improved their literature review, theoretical background presentation, the figures, and the relevant parameters and results discussion.